# Type of RNA Packed in VLPs Impacts IgG Class Switching—Implications for an Influenza Vaccine Design

**DOI:** 10.3390/vaccines7020047

**Published:** 2019-06-04

**Authors:** Ariane C. Gomes, E. S. Roesti, Aadil El-Turabi, Martin F. Bachmann

**Affiliations:** 1The Jenner Institute, Oxford University, Oxford OX3 7BN, UK; aadil.el-turabi@ndm.ox.ac.uk (A.E.-T.); martin.bachmann@me.com (M.F.B.); 2Department for BioMedical Research (DBMR), Inselspital, University of Bern, 3010 Bern, Switzerland; elisa.roesti@dkf.unibe.ch; 3Immunology, RIA, Inselspital, University of Bern, 3010 Bern, Switzerland; 4Department of Dermatology, University Hospital Zurich, 8952 Schlieren, Switzerland

**Keywords:** VLPs, RNA, TLR7, IgG

## Abstract

Nucleic acid packed within virus-like particles (VLPs) is shown to shape the immune response and to induce stronger B cell responses in different immunisation models. Here, using a VLP displaying the highly conserved extracellular domain of the M2 protein (M2e) from the influenza viruses as an antigen, we demonstrate that the type of RNA packaged into VLPs can alter the quality of the induced humoral response. By comparing prokaryotic RNA (pRNA), eukaryotic RNA (eRNA) and transfer RNA (tRNA), we find that pRNA induces the most protective IgG subclasses using a murine influenza model. We provide evidence that this process is predominantly dependent on endosomal Toll-like receptor (TLR7), and rule out a role for cytoplasmic mitochondrial antiviral signalling protein (MAVS) and its upstream retinoic acid-inducible gene-I-like receptors (RIG-I). Our findings provide considerations for the rational design of VLP-based vaccines and the immunomodulation exerted by TLR7 ligands packaged within the particles. Based on this work, we conclude that VLPs packing prokaryotic RNA must be preferred whenever a response dominated by IgG2 is desired, while eukaryotic RNA should be employed in order to induce a response dominated by IgG1.

## 1. Introduction

Virus-like particles (VLPs) are an attractive vaccine platform due to their excellent safety profile associated with high immunogenicity and relatively low production costs under good manufacturing practices (GMP) [1,2]. The high immunogenicity of these particles is due to virus-like pathogen associated molecular patterns (PAMPs) such as repetitive structure, size and, in some cases, the presence of nucleic acid (reviewed [3]). RNA derived from the host cell expression system is often co-packaged into VLPs derived from RNA bacteriophages as poly-anionic molecules are involved in the assembly and stability of the particle [4]. This is the case of widely used VLPs such as Qβ [5] and AP205 [6]. In addition to stability, the presence of nucleic acids within VLPs has been extensively shown to engage several pattern recognition receptors (PRRs) such as TLRs contributing to the immunogenicity of such particles [7,8,9,10].

The ability to recognise pathogens as foreign and potentially dangerous is an important feature of the immune system and it is promoted by different PRRs that orchestrate the immune response by triggering cytokine production, cell recruitment and maturation, and ultimately promoting inflammation or tolerance [11]. For B cells, an important role for TLRs has been established in the development of humoral responses, as demonstrated by the fact that total immunoglobulin G (IgG) levels are severely diminished and IgG2a induction is abolished in the Myd88-/- mice [12]. For T cell independent (TI) antigens such as VLPs, extensive literature shows that events of isotype switching are promoted by direct TLR engagement in B cells and that it cannot be rescued by TLR activation in dendritic cells (DCs), activation of T helper cells (T_H_) [12,13] or addition of T_H_1 cytokines such as IFNγ and IL-12 [14]. Instead, isotype switching to IgG2a/c and IgG2b is promoted by dual engagement of B cell receptor (BCR) and TLR in B cells following uptake of VLPs containing TLR ligands [10,13].

The type of RNA packed within bacteriophage derived VLPs depends entirely on the chosen expression system (e.g., *E. coli*, HEK cells, insect cells to name a few) and there is evidence that prokaryotic and eukaryotic RNAs of different classes (e.g., rRNA, mRNA and tRNA) albeit all being TLR7 agonists, have distinct capacities to activate and induce cytokine production in DCs in vitro [15,16]. However, the same has not been investigated in B cells and even less is known whether these observations would impact the protective immune response induced by vaccines. The distinct interactions with TLRs is attributed to the level of secondary modifications on RNA such as 2′-O-methylation of nucleotides [15]. It has been further noted that prokaryotic mRNA possesses less secondary modification, followed by eukaryotic RNA, while tRNA has the highest level of modifications. Not surprisingly the level of secondary modification was correlated to the strength of TLR-dependent activation of DCs. Considering the pivotal role of TLR7 signalling in B cells for efficacy of vaccines against influenza we wanted to evaluate the impact of different classes of RNA in the formulation of VLP-based vaccines.

The murine IgG family is composed of 4 major subclasses IgG1, IgG2a/c, IgG2b and IgG3. Of note, the IgH-1b haplotype of BALB/c mice includes the IGHG2A gene but not the IGHG2C gene, whereas C57BL/6 includes the IGHG2C gene but not the IGHG2A gene [17]. The different constant regions are implicated in distinct effector functions in the humoral responses and have been shown to be particularly important for the control of influenza A by vaccination [18] and by monoclonal antibodies. The importance of IgG subclasses has been demonstrated using M2e specific monoclonal antibodies with identical variable regions but fused to murine IgG1 or IgG2a. The IgG2a monoclonal antibodies were much more potent at protecting against mortality and morbidity in an influenza virus challenge, owing to the distinct effector interaction capacities of the Fc-γ portions of the IgG subclasses [18]. In addition, it has been demonstrated that isotype switching to the protective form IgG2 is strictly TLR dependent, and TLR dependent isotype switching to IgG2 was crucial for protection and reduction in morbidity in influenza vaccines based on M2e [18]. The same findings regarding the importance of TLR stimulation by influenza vaccines have been shown with other vaccine formulations employing HA as an antigenic target [19], further establishing that TLR stimulation is important for the control of the disease and not a requirement restricted to VLP-based vaccines [19].

M2 is a homo-tetramer integral membrane protein and its extracellular domain (M2e) is highly conserved across most Influenza A strains remaining relatively unchanged throughout the major influenza pandemics. For this reason, M2e may be an antigenic alternative for a vaccine with broader spectrum of protection against different influenza strains [20]. M2e is a poorly immunogenic protein during natural infection and antibodies raised by whole virus vaccines are mostly dominated by anti-hemagglutinin (HA) and anti-neuraminidase (NA) antibodies, whilst anti-M2e antibodies remain largely absent. However, in contrast to conventional vaccines and natural infection, active immunisation employing subunit vaccines based on M2e as antigen has been shown to confer cross-protection when administered in combination with adjuvants in murine, ferret and macaque models [18,21,22]. It has been further demonstrated in different vaccination models employing M2e that protection is mediated not through direct viral neutralisation, but rather through Fc-γ receptor interaction of M2e-specific antibodies with innate immune cells. Specifically, morbidity was greatly reduced in responses dominated by IgG2 Fc portion even when the antibody variable region was identical [23,24]. Therefore, the quality of the humoral responses (e.g., dominant IgG subclasses) and not only the magnitude of the response induced by M2e-vaccines may be crucial to providing protection [18].

In the present study, we demonstrate that not only the presence of the TLR7 ligand (i.e., RNA) is important to promote isotype switching to IgG2 but also the level of secondary modifications of the TLR ligand employed. We observed that humoral responses elicited against a VLP-based vaccine containing RNA of different sources (prokaryotic and eukaryotic mRNA and tRNA) induced distinct IgG subclass-distributions in a strictly TLR7-dependent fashion, while the cytoplasmic adaptor protein MAVS and its upstream receptors were not involved in the process. The results presented here support M2e-based vaccines as a highly efficacious influenza vaccine. Moreover, we further elucidate the versatile nature of VLPs as a vaccine platform able to rationally modulate and skew the humoral immune response by choosing appropriate expression systems.

## 2. Material & Methods

### 2.1. Data Availability

The datasets generated and/or analysed during the current study are available from the corresponding author on reasonable request.

### 2.2. Qβ Preparation

Purification—VLP derived from the coat protein of the bacteriophage Qβ were recombinantly expressed in *E. coli* and purified as previously described [6].

Disassembly—Purified Qβ VLPs were disassembled by addition of the reducing agent 1,4-Dithioerithrol and precipitation of packed RNA. The dimeric Qβ coat protein was purified with a cation exchange SP-Sepharose FF column and size exclusion in a Sephacryl S-100 HR column (both from GE Healthcare, Little Chalfont, UK).

Re-assembling Qβ VLPs—Purified coat protein in reducing conditions were mixed with RNA and H_2_O_2_ was added to induce the formation of disulphide bonds. Re-assembling was assessed by agarose gel, dynamic light scattering (DLS) and electron microscopy (EM).

Vaccine preparation and peptide sequence—M2e peptide SLLTEVETPIRNEWGC-azide were coupled to Qβ VLPs through the copper-free click chemistry crosslinker DBCO following instructions of the product (Merck (Sigma), Gillingham, UK).

### 2.3. Electron Microscopy and Dynamic Light Scattering

Suspensions of VLPs were adsorbed on carbon-formvar coated copper grids, 400 mesh (Agar Scientific, Stansted, UK) and stained with 1% uranyl acetate (pH 6.8). The grids were examined with a Tecnai T12 microscope. VLP samples were diluted to 0.3 mg/mL in 20mM HEPES buffer and analysed at room temperature with a Malvern Zetasizer Nano ZS (Malvern Panalytical, Malvern, UK) equipped with a He-Ne laser (633 nm). Data analysis was conducted with DTS software (version 4.2), using a non-negatively constrained least squares (NNLS) fitting algorithm. Dispersant refractive index and viscosity of the dispersant were assumed to be 1.33 and 1.02 cP, respectively.

### 2.4. Lipopolysaccharide (LPS) and RNA Content of Preparations

Final preparations of the vaccine were tested for endotoxin content using the Pierce LAL chromogenic LPS test (Thermo Fisher Scientific, Paisley, UK) following the manufacturer’s instructions.

RNA was measured by densitometry on an agarose gel stained with SYBR safe and confirmed by Quant-It RNA HS kit and Qubit fluorimeter (Thermo Fisher Scientific) according to manufacturer’s instructions.

### 2.5. Animals and Immunisations

All mice used in this study were bred in specific pathogen-free (SPF) conditions. This study was carried out in accordance with the recommendations of the Animals (Scientific Procedures) Act 1986 (ASPA) and European Directive 2010/63/EU on the protection of animals used for scientific purposes and in accordance with the Swiss Animals Act (455.109.1) (September 2008, 5th, University of Bern). Wild-type mice were purchased from Envigo (Huntingdon, UK).

The C57BL/6 *MAVS^−/−^* (also known as *Cardif^−/−^)* mice were kindly provided by John Cambier and have been described elsewhere [25]. The TLR7*^−/−^* (B6.129P2-Tlr7tm1Aki) mouse strain was a kind gift from Dr. Pål Johansen.

For antibody responses, mice were immunised s.c. with 50 μg of Qβ(pRNA)-M2e, Qβ(eRNA)-M2e, or Qβ(tRNA)-M2e and serum collected from the lateral tail vein. Days of sample collection are indicated in each experiment.

### 2.6. ELISA

M2e-specific antibodies were measured by coating plates with the M2e peptide crosslinked to the carrier protein RNAse. Peptide and carrier protein were crosslinked using SMPH (Thermo Fisher Scientific) following the manufacturer’s instructions. VLP-specific antibodies were measured by coating Nunc Immunoplates (Thermo Fisher Scientific) with Qβ VLPs. Immune serum was serially diluted 12x (2-fold dilution) starting at 1:50 for IgG1 and 1:100 for IgG, IgG2b and IgG2c. Subtypes were detected with HRP-conjugated goat anti-mouse IgG1, IgG2b and IgG2c antibodies (all from Thermo Fisher Scientific).

### 2.7. Type I IFN Response Measurement

ELISA—Serum from immunised mice was tested following manufacturer’s instructions. The ELISA Kit was purchased from Thermo Fisher with the sensitivity range of 31–2000 pg/mL.

IFIT induction on 3T3 cells—Mouse type I IFN levels in the serum was measured by incubating 3T3 cells seeded with 25 μL of serum from immunised mice. Cells were incubated for 3 h with serum. Following incubation period, cells were immediately prepared for RNA extraction.

### 2.8. qPCRs

Total RNA was isolated using the RNAeasy Mini Kit (Qiagen, Manchester, UK) with in column DNA digestion. cDNA and non-RT controls were synthesised using the iScript cDNA Synthesis kit (BioRad, Hemel Hempstead, UK). Quantitative RT-qPCR was performed in duplicate using SYBR Green Master Mix (BioRad) on an CFX connect machine (BioRad). Results were analysed with a CFX Maestro Software (BioRad) employing the comparative Ct methods and GAPDH as the endogenous normalisation control. Sequences are described on Table 1.

### 2.9. Cytometry

VLP uptake—Qβ VLPs were labelled with AlexaFluor 647 or PE containing the NHS reactive group as instructed by the manufacturer (all from Thermo Fisher Scientific). C57BL/6 mice (9–12 weeks old; Envigo) were injected s.c. with 30 μg of fluorescent VLP and peptide into the hind leg.

Popliteal and inguinal LNs were isolated and a single-cell suspension was prepared and stained for cell-specific markers. The following fluorochrome-labelled antibodies were used: anti-CD11c, anti-CD317 (all from eBioscience, Thermo Fisher Scientific), anti-TLR7, anti-CD69 and anti-F4/80, Live/Dead Aqua cell stain (all from Life Technologies, Thermo Fisher Scientific).

### 2.10. Influenza Challenge

Adult (eight-to-twelve week old) BALB/c female mice were immunised s.c. on days 0 and 14 with 50 μg with either Qβ(pRNA)-M2e or Qβ(eRNA)-M2e or as a control with 50 μg of Qβ. On day 21, mice were bled in order to determine anti-M2e IgG or subclass antibody titres by ELISA. Mice were then challenged at the indicated time point with a lethal dose (2 × LD50) of mouse-adapted influenza A virus PR8 (H1N1 strain was kindly provided by Dr. Manfred Kopf, ETH Zurich). Following the challenge, body weight and temperature were monitored daily at the same time of the day. Experimental end-point was determined based on body temperature reaching 33◦C or loss of 30% of their initial body weight, which is when the mice were euthanised.

Aiming to reduce the number of animals used in this study and the potential suffering imposed upon them, we made the decision not to include VLPs containing tRNA in the challenge, as the overall IgG titres were lower and specially owing to the similar IgG2/IgG1 ratio to VLP containing eukaryotic RNA (eRNA). Considering that our primary null hypothesis was that the ratio of IgG2/IgG1 does not affect the protection conferred by the vaccine, it was reasoned that Qβ(eRNA)-M2e versus Qβ(pRNA)-M2e would provide enough evidence to assess the validity of the hypothesis.

### 2.11. Statistical Analysis

Antibody titres were represented as OD50 (dilutions that reached half the maximal OD observed for the assay). Data was examined by ANOVA (whenever ANOVA F statistics were significant, *p* < 0.05) followed by multiple T test corrected by Bonferroni correction. Differences were considered statistically significant at *p* < 0.03. Survival rate was analysed by Log-rank (Mantel–Cox) test.

## 3. Results

### 3.1. Eukaryotic mRNA Fails to Promote Isotype Switching to the Same Level as Prokaryotic mRNA If Packaged into VPLs

VLPs and other classes of nanoparticles induce a humoral response dominated by IgG1 in the absence of packed RNA or DNA in mice [14,22]. In contrast, in the presence of ssRNA packaged during production in *E. coli*, the humoral immune responses are dominated by the IgG2 subclasses. To assess the capacity of other classes of RNA to promote IgG2-skewed responses, we generated VLPs derived from the bacteriophage Qβ packing prokaryotic RNA from *E. coli*, eukaryotic RNA from the cell line HEK 293T, and yeast derived tRNA by in vitro re-assembly of the VLPs. The integrity of VLPs was assessed by electron microscopy, dynamic light scattering (DLS) and native agarose gels, demonstrating that Qβ can be re-assembled with the three different classes of RNA. All VLPs had similar shape and diameters of approximately 30 nm as measured by EM (Figure 1A), and were 30 to 80 nm in size as measured by DLS, and gave similar negative net charge as measured by ζ potential (Figure 1D). Moreover, similar amounts of RNA were packaged regardless of the class of VLP (Figure 1B) as assessed by the charge-dependent migration pattern in a native agarose gel. The three preparations were coupled to the M2e peptide derived from the extracellular domain of the conserved M2 protein from Influenza A [20]. The three VLPs are henceforth represented as Qβ(pRNA) for VLPs packaging prokaryotic RNA, Qβ(eRNA) for eukaryotic RNA and Qβ(tRNA) for eukaryotic transfer RNA (tRNA). To rule out additional TLR stimulation lipopolysaccharides (LPS) was measured. All the formulations had similar LPS concentration lower than 0.5 EU per dose of vaccine (Figure 1D), excluding LPS as variable affecting the relevant immune responses.

To evaluate the impact of the packaged RNA on humoral responses, C57BL/6 female mice were immunised s.c. with 50 μg of the respective Qβ VLPs and the total IgG and IgG subclasses (IgG1, IgG2b, IgG2c) against VLP and M2e peptide were assessed 10 days later by ELISA (Figure 2). VLPs containing different classes of RNA induced distinct humoral responses with different ratios of IgG subclasses. Likewise, the type of response against the peptides coupled to the VLPs followed the same pattern (Figure 2B). Qβ(eRNA)-M2e and Qβ(pRNA)-M2e immunised mice had similar circulating levels of total IgG, while Qβ(tRNA)-M2e showed an approximately 10-fold reduction in the circulating total IgG titres. To further characterise humoral responses, IgG1, IgG2b and IgG2c levels specific for the VLP as well as M2e were measured next. In accordance with the significantly lower total IgG levels, Qβ(tRNA)-M2e induced lower levels of all subclasses in comparison to the other groups, while Qβ(eRNA)-M2e and Qβ(pRNA)-M2e induced similar levels of IgG2b/c but significantly different levels of IgG1 (*p* < 0.03). Moreover, the ratio IgG2b/IgG1 was 3 times higher in the Qβ(pRNA)-M2e group compared to both Qβ(eRNA)-M2e and Qβ(tRNA)-M2e demonstrating the superior capacity of prokaryotic RNA at promoting the isotype switching to IgG2 over IgG1.

### 3.2. Isotype Switching Promoted by VLPs Is Independent of MAVS

Previous literature shows that RNA promotes isotype switching promoted by direct TLR7 engagement in B cells. Recent literature, however, suggests an additional role for cytosolic RNA sensors in the modulation of the humoral response [26]. Secondary modifications in the RNA is a prominent feature distinguishing prokaryotic from eukaryotic RNA recognized by the cytoplasmic RNA sensor RIG-I. To investigate a possible role for RIG-I in driving IgG2 responses, sex and age matched mice lacking the RIG-I downstream adaptor molecule MAVS were immunised following the same regimen and the IgG subtypes were evaluated 10 days after immunisation (Figure 3). For all the parameters evaluated, MAVS knockout had similar levels of total IgG, IgG1 and IgG2b/c, ruling out a role for the RIG-I/MAVS pathway in regulating isotype switching mediated by prokaryotic RNA in VLPs.

### 3.3. Activation of DCs and B Cells

Although isotype switching is dependent on direct TLR7 engagement in B cells [14], TLR7 and MyD88 expression in naive B cells have been shown to be modulated by IFN-α produced by pDCs [27]. Considering that DCs also respond differently according to the class of RNA, we wanted to assess whether DCs responded differently to the distinct VLP preparations.

First, to confirm the binding and/or uptake of Qβ VLPs by different subsets of DCs and B cells, mice were injected with VLPs coupled to a fluorescent label. After 24h, draining lymph nodes were processed and immune cells analysed for VLP uptake (Figure 4). Four percent of total macrophages (CD11c+F4/80+) were positive for the VLPs. Similar percentages of pDCs (CD11c+CD317+) and cDCs (CD11c+F4/80− CD317−) had taken up the VLP 20 h post immunisation. Up to 1% of B cells were positive for the VLP confirming the capacity of the VLP to interact with the different subsets of DCs and B cells.

To test the early immune response mediated by DCs and B cells, mice were immunised with 50μg of Qβ(pRNA), Qβ(eRNA) and Qβ(tRNA) s.c., and 20 h later, serum was collected to evaluate circulating levels of IFN-α. Serum levels of IFN-α were first assessed by commercially available ELISA kits; however IFN-α levels were below the detection limit of the assay (30 pg/mL). Considering that nucleic acid containing VLPs are shown to induce IFN-α production in vitro, a more sensitive test to determine type I IFN responses was employed. To this end, 3T3 cells were incubated with serum from immunised mice and the levels of IFIT1 gene, an interferon-stimulated genes (ISG) was measured by qPCR. Serum of immunised mice induced different levels of IFIT1 gene depending on the type of vaccine received (Figure 4B), with Qβ VLP containing prokaryotic RNA inducing comparatively higher levels of the expression of the IFIT1 gene on 3T3 cells.

Next, we wanted to evaluate whether the type of RNA was regulating key markers of the immune response. TLR7 and CD69, two receptors that are up-regulated in response to IFN-α were measured on splenic B cells 20 h following immunisation in order to analyse the activation state of B cell in vivo. We found that TLR7 was up-regulated in B cells from immunised mice (Figure 4C) when compared to naïve mice, but there were virtually no differences in the expression of TLR7 between Qβ(pRNA) compared to Qβ(eRNA) (Figure 4D) despite strikingly different levels of IFIT1 induction. CD69 expression was higher on B cells from mice immunised with Qβ(pRNA) and the increase was TLR7-dependent, as seen by the reduced expression of CD69 on B cells from immunised TLR7 KO mice.

### 3.4. Protection against Challenge with Influenza Virus

Following assessment of humoral immune responses, the influence of the IgG subclasses composition in influenza challenge was tested. To that end, mice immunised in a prime-boost regimen (day 0 and 14) with Qβ VLPs coupled to M2e packing either prokaryotic RNA or eukaryotic RNA were challenged on day 28 later (Figure 5A) with the previously determined lethal dose (2 × LD50) of the mouse-adapted influenza A virus PR8 (H1N1 strain). Temperature and weight were monitored daily until control group reached the severity limit of 30% weight loss or body temperature lower than 33 °C, when mice were euthanised. All the procedures conformed to local ethical frameworks in place aiming for the reduction of suffering and numbers of animals used. Mice immunised with both formulations survived the lethal dose of influenza, but there were observable differences in morbidity. Mice immunised with Qβ(pRNA)-M2e showed reduced weight loss during the challenge and better temperature control when compared to Qβ(eRNA)-M2e.

Humoral responses in challenged mice were assessed following immunisation and challenge (Figure 6). As expected, antibody titres were greatly boosted by a second dose of the vaccines but the ratio of IgG2/IgG1 remained the same (Figure 6D). Interestingly, M2e-specific humoral responses were not detected in the mock group even post challenge, confirming that influenza virus infection does not normally trigger an anti-M2 response [20]. In contrast, infection boosted M2e-specific antibody levels in immunised animals, indicating that natural infection may be able to maintain M2e-specific antibody levels in immunised individuals [18].

## 4. Discussion

Previous data has extensively shown that the presence of nucleic acid packed within VLP drives isotype switching to IgG2 and this process is a B-cell intrinsic, Myd88-dependent process [8,14,18]. In the present study, we provide further evidence that not only does the presence of RNA within VLPs drive the magnitude and quality of immune responses, but also that the type of RNA (i.e*.,* prokaryotic, eukaryotic, transporter) influences the dominant IgG subclass induced. This is an important consideration for vaccine development considering that the different IgG subclasses have distinct effector capabilities and that this can greatly influence immunogenicity of the particle.

During process development, the choice of expression system usually revolves around yield, cost and sometimes the convenience of familiarity with the system and techniques in place. This strategy, however, usually fails to evaluate the impact of the production process on the immunogenicity of the particle. For instance, the most common platforms for protein production are *E. coli* or eukaryotic systems such as insect cells, yeast or mammalian cells. However, the secondary modifications in the nucleic acid of each of the expression system vary greatly [16,28] and now we have shown for the first time to our knowledge that the type of RNA packed into the VLPs impacts humoral responses. This has been demonstrated by the increased levels of IgG2b and IgG2c and reduced levels of IgG1 in mice receiving Qβ(pRNA), also evidenced by the ratio of the two subclasses. Furthermore, we found no role for MAVS in the antibody response elicited by VLPs in disagreement to the findings described by Zeng et al. [26]. Although we do not rule out a role for such receptors in isotype switching as a process, our data argue that in the context of VLP-mediated TLR stimulation, receptors such as RIG-I may not be essential to contribute to isotype switching or protection.

Following the conclusion that Qβ(pRNA) induces higher levels of IgG2b/c in detriment of IgG1 compared to Qβ(eRNA) and Qβ(tRNA), and that this process is not dependent on MAVS/RIG-I, we attempted to identify activation markers in B cells that could be related to dominance of IgG2b/c in the Qβ(pRNA) responses. To that end, we measured the expression of the activation molecule CD69 in B cells following immunisation, showing that CD69 expression is higher in mice receiving Qβ(pRNA) compared to the other formulations, and that this CD69 up-regulation was abolished on a TLR7−/− knockout mice. We also measure t-bet expression—a transcription factor known to be involved in isotype switching [29] but found no differences (not shown). RNAs are well known to induce IFN-α production and this cytokine is known to play a role in modulating TLR7 and MyD88 expression in B cells in vitro. In the present study, however, we were unable to replicate such findings. Even though Qβ(pRNA) was superior at inducing type I IFN responses in vivo*,* these differences did not translate to differences in TLR7 expression in B cells. Recent literature also suggests a role for type III IFN produced by NK cells in modulating TLR7 sensitivity and expression by B cells [30]; however, this was not addressed directly in this study. Finally, we have demonstrated that mice that exhibited a higher IgG2/IgG1 ratio had significantly reduced morbidity.

An example of how the findings of this study may be translated into clinical solutions can be illustrated by the VLP-based HPV vaccines. All three currently available prophylactic HPV vaccines do not contain packaged nucleic acid in their final formulation, but rather benefit from the inclusion of adjuvants including some engaging TLR4 [31]. This supports the role of TLR in driving immunogenicity and efficacy of vaccines. Taking our findings into consideration, it is not unreasonable to speculate that in other indications, use of VLPs packaging RNA and expressed in prokaryotic systems would be a more efficient and effective platform because they induce the IgG subclass of higher effector function, eliminating the need to include additional adjuvants that may result in increased costs and unforeseen side-effects for the vaccine. Collectively, our data support the growing trend of employing ligands for PRRs as adjuvants and provides further evidence to optimise the design of adjuvants of this kind. We demonstrate here that even though both vaccines (i.e., containing pRNA or eRNA) perform similarly in terms of protection against lethal infection, there were significant differences in the relative morbidity experienced during the acute phase and recovery from the disease. The presented evidence indicating that the type of RNA influences the protective capacity of M2e based vaccines may therefore contribute to the rational design of vaccines in the future.

In conclusion, our results demonstrate in vivo that manufacturing differences resulting in distinct types of packaged RNA in VLPs, has an impact on the immunogenicity of the vaccine. We propose that during the development phase of VLPs, the type and quality of RNA packed should be assessed and thoroughly considered while evaluating which VLP should be employed as vaccine platform. Furthermore, when expressing VLPs in either prokaryotic or eukaryotic cells, RNA may be used as means of modulating the immune response to achieve the desired outcome. As in certain situations antibodies with increased or reduced effector capacity may be desired, depending on the disease and target structure, choosing the right expression system and RNA quality may be an important point of consideration.

## Figures and Tables

**Figure 1 vaccines-07-00047-f001:**
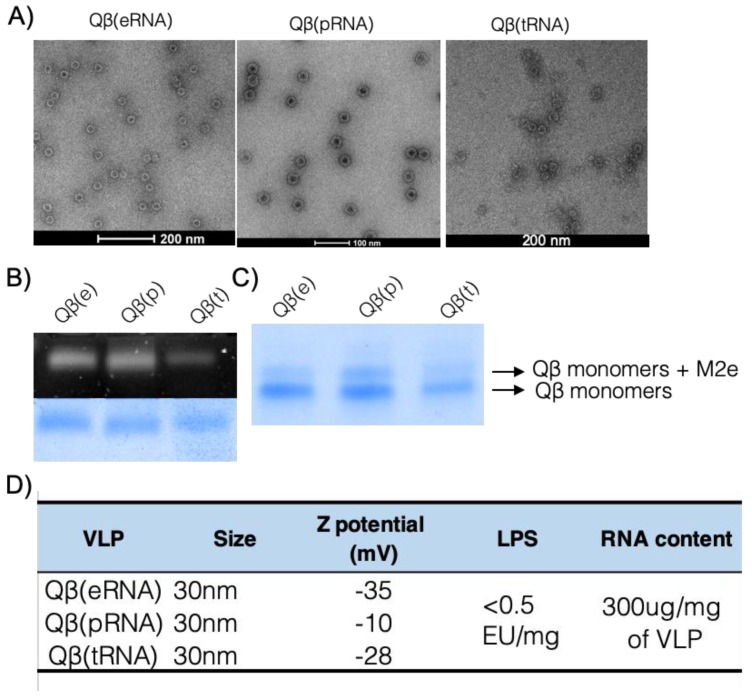
Vaccine preparation and characterisation. (**A**) Electron micrograph of re-assembled VLPs. Qβ VLP containing eukaryotic RNA (eRNA), prokaryotic RNA (pRNA) and tRNA (tRNA) have similar structure and size. (**B**) Native analysis of RNA and protein content of re-assembled VLPs. Upper panel showing SYBR staining and bottom panel showing protein staining of same agarose gel. (**C**) Similar coupling efficiency of different VLPs and M2 peptides on a reduced SDS-PAGE gel electrophoresis. (**D**) Characterisation of VLPs. Size was taken as an average of values obtained in the EM images. Z potential was measured by DLS and is expressed as mV. LPS content expressed on endotoxin units (EU) per milligram of VLP. Average RNA content per mg of VLP. Results are representative of more than 5 independent purification and re-assembly studies.

**Figure 2 vaccines-07-00047-f002:**
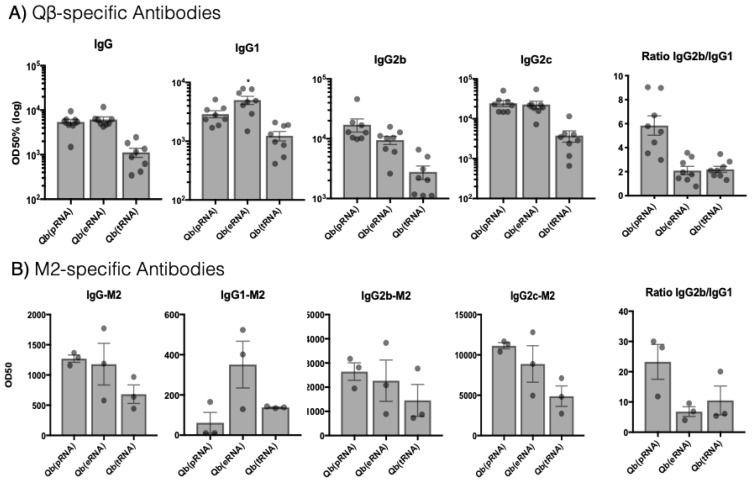
Total IgG and subtypes and IgG2/IgG1 ratio are influenced by the type of RNA packaged by virus-like particles (VLPs). IgG, IgG1, IgG2b and IgG2c 10 days after s.c. immunisation with 50 μg of VLP. Qβ(eRNA)-M2e induces significantly higher levels of antigen-specific IgG1 compared to the two other vaccines (*p* > 0.03) against VLP and M2. Ratios of IgG2b:IgG1 are represented as isotype geometric mean titres. (**A**) VLP-specific antibodies. Mean + SEM n = 8 mice per group (**B**) M2-specific antibodies. Mean + SEM n = 3 mice per group. OD50 with titres defined as those dilutions that reached half the maximal OD observed for the assay. Mean ratio of IgG2b/IgG1 + SEM. Results are representative of three independent experiments.

**Figure 3 vaccines-07-00047-f003:**
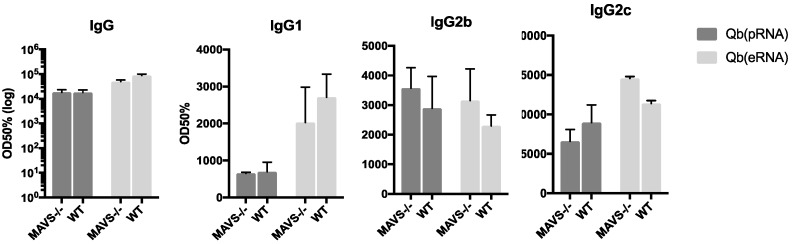
Isotype switching promoted by VLPs is independent of MAVS. Total IgG, IgG1, IgG2b and IgG2c were measured by ELISA 10 days after s.c immunisation with 50 μg of VLP in age and sex-matched MAVS knock-out and WT counterparts. VLP-specific antibodies. Mean + SEM n = 5. OD50 are titres defined as those dilutions that reached half the maximal OD observed for the assay.

**Figure 4 vaccines-07-00047-f004:**
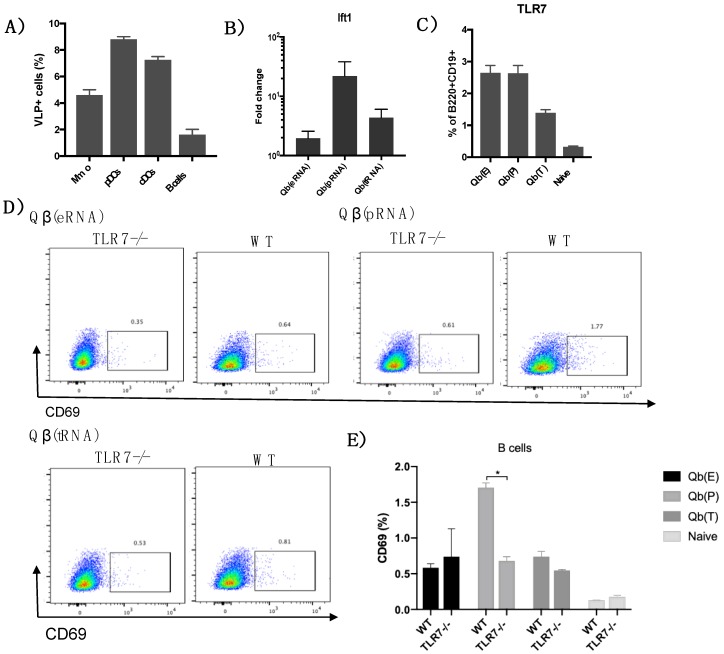
TLR7 dependent immune response of B cells and DCs. (**A**) Axis indicates the percentage of macrophages (Mmo), pDCs, cDCs and B cells positive for *E. coli*-expressed Qβ 24 h post injection. (**B**) Induction of the ift1 gene on 3T3 cells treated with serum of mice immunised with VLPs loaded with different RNAs 20 h post immunisation, measured by RT-qPCR. Data are presented as fold changes compared to the average cells treated with sera of naïve mice. (**C**) TLR7 expression in B cells (B220^+^CD19^+^) from immunised mice 24 h post immunisation. (**D**) CD69 expression on splenic B cells 24 h post immunisation of WT and TLR7 knock out mice. Gated on B cells (B220^+^CD19^+^). (**E**) Bar plot of CD69 expression on B cells 24 h post immunisation. Data represented as mean + SEM n = 3 mice. * *p* < 0.04. Results are representative of 2 independent experiments.

**Figure 5 vaccines-07-00047-f005:**
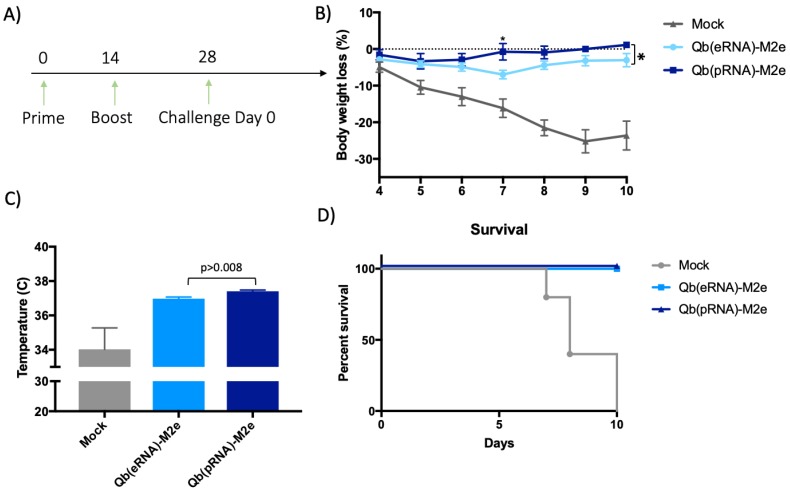
Reduced morbidity symptoms of mice immunised with formulations containing prokaryotic RNA in an influenza challenge. (**A**) Immunisation and challenge schedule. 8–12 weeks female BALB/c mice were immunised s.c. with 50 μg of Qβ(pRNA)-M2e or Qβ(eRNA)-M2e, or PBS as a control on day 0 and 14. Mice were then challenged with a lethal dose of the mouse-adapted influenza A/PR/8/34 virus (2 × LD50) on day 28. (**B**) Body weight (%) represented as the mean +/− SEM percentage of weight loss (n = 5 mice). Statistical significance measured by Welch’s T test (*p* < 0.007). (**C**) Average temperature from day 4 to 11 post challenge. mean +/− SEM (n = 5 mice). Statistical significance measured by Welch’s T test (*p* < 0.008). (**D**) Percentage of survival of immunised and control groups following challenge. *****: One mouse from the Qβ(pRNA) group was prematurely culled due to unrelated sickness. Mice that reached the limit severity score were euthanised and the last observed parameter was used for the following days and average calculations (Last observation carried forward, LOCF).

**Figure 6 vaccines-07-00047-f006:**
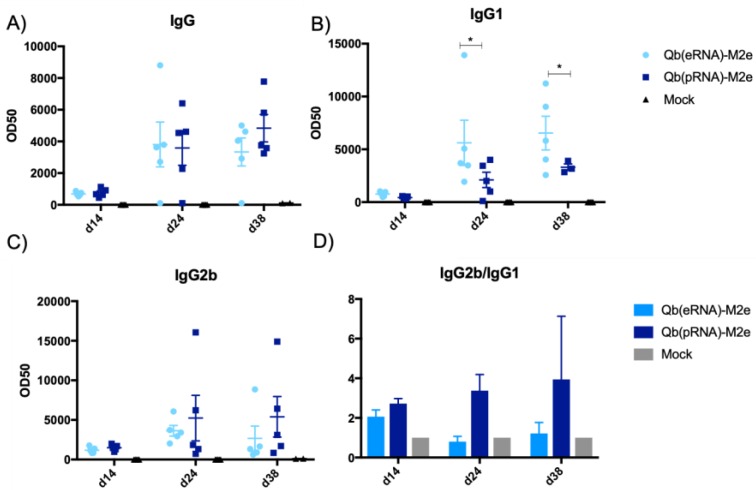
Humoral responses against M2e. Circulating levels of IgG, IgG1 and IgG2b measured on day 14, day 24 and day 38. Vaccine dose was 50 μg of VLP, injected s.c. Qβ(eRNA)-M2e induces significantly higher levels of antigen-specific IgG1 compared to the other vaccines (* *p* > 0.04) against VLP and M2. Ratios of IgG2b:IgG1 are represented as the ratio of the isotype geometric mean titres (OD50). Antibody data represented as mean + SEM, n = 5. Ratio represented by mean ratio of IgG2b/IgG1 + SEM. N = 5.

**Table 1 vaccines-07-00047-t001:** Primer sequence.

Gene	Forward		Reverse	Supplier
IFIT1	5′AGAACAGCTACCACCTTTAC		5′TTCTTGATGTCAAGGAACTG	Sigma
GAPDH	qMmuCEP0039581 (Assay ID)		sequence not provided by the supplier	BioRad

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
