# Peer review of "Type of RNA Packed in VLPs Impacts IgG Class Switching—Implications for an Influenza Vaccine Design"

_vaccines, 2019, doi:10.3390/vaccines7020047_

Round 1
Reviewer 1 Report
The text is a bit too long and could be/should be written in a more concise way. This is especially true for thé Matérial and Methods section and for thé légends to thé figures.
This article is most interesting and certainly warrants publication. Its conclusions seem to be well established.
However,the paper would certainly gain interest if was demonstrated that IgG2b are more effective than IgG1 at protecting against influenza and/or other diseases. In other words, has it been found that vaccines which elicit IgG2b are more protective (or less protective) than those which elicit IgG1? (See also Discussion, lines 396-99). This might actually also be of great interest for passive immunization.
With regard to writing, the text is a bit too long. Also, several corrections are needed, such as:
-Abstract, line 16: tRNA is 'transfer RNA', not 'transport RNA'!
-p 8, line 237: «body weight and temperature of (...of what?) was were monitored daily»
-p 9, line 244: «between VLP containing eRNA» ...and what? (between A ...and B)
-p 9, line 260: «faisl» should be «fails»
-p 13, line 369: «mice immunized with both either formulations»
-p 13, line 383: «sepcific»should be «specific»
-p 15, line 425: «was abolished on a TLR7-/-» ...on a TLR7- what? (background? Host? Mouse? )
Most legends to the figures are a bit too long (too wordy).
There also are too many abbreviations with no explanation such as: BCR (l 54); IGHG2Aand IGHG2C (l 73-74), MAVS (l 114 and 174), DBCO (l 146), DTS (l 154), LAI and LPS (l 161), SYBR (l 163), SMHP (l 184), IFIT (l 197), GAPDH (l 209), PE and NHS (l 216), ANOVA (l 253), etc..
What is the «ISG IF1T1 gene»? (l 345). What are «ligands for PRRs»? (or what are PRRs?) (l 447).
With regard to the figures, why are they two blocks in Fig 5? Meaning of the lower block??
Author Response
1) We thank the reviewer for pertinent comments and suggestions. The document has been revised and shortened aiming to turn the text more concise.
2) Yes, this has been demonstrated in a influenza model. Ref #23 and #24. Lines 107-108 were re-drafted to make it more clear.
3) Addressed.
4) Relevant abbreviations added.
5) Same figure, the gap is to keep the figure proportional.

Reviewer 2 Report
Vaccines (ISSN 2076-393X)
Manuscript ID - vaccines-508117
This is a review of the following paper:
Type of RNA packed in VLPs impacts IgG class switching – Implications for an influenza vaccine design
Authors: Ariane C. Gomes, Elisa Roesti , Aadil El-Turabi , Martin Bachmann
Specific Comments:
This paper is very well prepared and therefore easy to review.
The experimental approach and descriptions of the steps is very clear and that makes evaluation of the data straight forward. All sections are well written.
Two suggestions:
1. The ELISA data demonstrate relative differences in approximate, relative magnitude of the antibody responses. Comparing the numbers of total IgG and the isotypes can be misleading. Is there any way to clarify the magnitude of the antibody responses? Perhaps relative to a known set of control/standard sera?
2. The clinical signs observed in the challenge of immunity studies are very interesting. This deserves more emphasis in conclusions and discussion. Also this provides a potential strong link to clinical vaccinology for other species (including humans)
Author Response
We thank the reviewer for pertinent comments and suggestions. The document has been revised and suggestions amended.
1) The levels of each subclass are not compared to other subclasses as this is indeed a misleading and artificial number. The comparisons are made only for a given subclass across the groups, which can be accurately measured by OD50.
2) Differences observed on morbidity were added to the discussion.
